# Cognitive aptitude, peers, and trajectories of marijuana use from adolescence through young adulthood

**Brian C. Kelly** [1] [ORCID]*, **Mike Vuolo** [2]

**1** Purdue University, Dept. of Sociology, West Lafayette, Indiana, United States of America, **2** The Ohio State University, Dept. of Sociology, Columbus, Ohio, United States of America

☯ These authors contributed equally to this work.
* bckelly@purdue.edu

**Data Availability Statement:** The data are publicly available data from the National Longitudinal Survey of Youth 1997. The data are overseen by the U.S. Bureau of Labor Statistics: https://www.bls.gov/nls/nlsy97.htm.

## Abstract

### Background

Using a nationally representative longitudinal cohort, we examine how cognitive aptitude in early adolescence is associated with heterogeneous pathways of marijuana use from age sixteen through young adulthood. We also examine whether this relationship can be explained by the role of cognitive aptitude in the social organization of peer group deviance.

### Methods

Using the National Longitudinal Survey of Youth 1997, we identified 5 latent trajectories of frequency of marijuana use between ages 16 and 26: abstainers, dabblers, early heavy quitters, consistent users, and persistent heavy users. Multinomial regression assessed the relationship of cognitive aptitude in early adolescence with these latent trajectories, including the role of peer group substance use in this relationship.

### Results

A one decile increase in cognitive aptitude in early adolescence is associated with greater relative risk of the dabbler trajectory (RR = 1.048; p < .001) and consistent user trajectory (RR = 1.126; p < .001), but lower relative risk of the early heavy quitter trajectory (RR = 0.917; p < .05) in comparison with the abstainer trajectory. There was no effect for the persistent heavy user trajectory. The inclusion of peer group substance use–either via illegal drugs or smoking–had no effect on these relationships.

### Conclusions

Adolescents who rate higher in cognitive aptitude during early adolescence may be more likely to enter into consistent but not extreme trajectories of marijuana use as they age into young adulthood. Cognition may not influence patterns of marijuana use over time via the organization of peer groups.

**Funding:** The authors received no specific funding for this work.

**Competing interests:** The authors have declared that no competing interests exist.

# Introduction

Major policy changes for medical and recreational marijuana over the past two decades have refocused attention on the effects of cannabis use, particularly with respect to social and cognitive development among young people. A key focus for this line of inquiry has been on the potential for marijuana use to impair cognition during an important period of human development. The transition from adolescence to young adulthood is typically a period of heightened substance use behaviors [1], making the relationship of substance use to cognition particularly acute during this time. Furthermore, research contends that young people perceive marijuana use as highly normalized and with relatively few health and social risks [2]. Indeed, across several generations, a majority of American young adults have tried marijuana at least once by the time they have reached age 25 [3]. Given this widespread use of cannabis, the possibility of acute and long-term effects on cognition remains an important area of inquiry.

Considerable research has focused on the influence of marijuana use on cognition, academic performance, and educational attainment. Research often highlights the possibility of a range of harmful outcomes for individual and social well-being related to cognitive aptitude and decision making, including lower educational attainment, reduced employment, impaired mental health, and criminal behavior (e.g. 4–6], among other possible adverse outcomes of marijuana use among young people [7]. While the focus on the possibility of adverse effects has been considerable within the literature, less attention has been placed on the role of cognition-based selection processes into patterns of substance use.

We aim to contribute to the growing literature on the relationship between cannabis use and cognition through an analysis of how cognitive aptitude in early adolescence may shape trajectories of marijuana use from adolescence through young adulthood. Using a nationally representative longitudinal cohort, we flip the widely studied pathway of marijuana use influencing cognitive aptitude on its head by examining how cognition in early adolescence is associated with heterogeneous pathways of marijuana use from age sixteen through young adulthood. Importantly, we examine whether the relationship between cognitive aptitude and marijuana use can be explained by the role of cognitive aptitude in the social organization of peer relationships among young people. In this manner, we consider whether cognition may contribute to social processes rather than simply function as an internal psychological trait.

## Marijuana use and the life course

Marijuana is most frequently used by Americans during the period in which they make the transition from adolescence to adulthood [3]. This transition has been characterized as a period during which a wide range of risk behaviors emerge, substance use being only one dimension of risk taking [1]. The high prevalence of substance use at this point in the life course has been attributed to the major changes in social roles and responsibilities that co-occur. The extended duration of the shift from adolescence to adulthood during the 21st century has also engendered a period of liminality that provides freedoms that facilitate substance use.

Examinations of this transitional period indicate that substance use may shape how young people traverse from adolescence to adulthood. Prior research has indicated that marijuana use is associated with several poor outcomes in adulthood related to personal and social well-being, including adverse impacts on educational attainment, employment, health, mental health, and criminal behavior, among other outcomes [e.g. 4–6, 8]. However, these assessments often primarily focused simply on whether or not an individual had ever used marijuana or assessing total use during adolescence without accounting for continued use or cessation in young adulthood. More recent research that accounts for the complexity of

differing frequency and severity of use over time has produced mixed results [9–14]. Additionally, recent work indicates that social and economic adversity only emerge for young people who engage in the heaviest patterns of cannabis use over an extended period of time [15]. Yet, it remains unclear whether cognitive aptitude prior to the transition to adulthood may shape how marijuana use unfolds prior to later life outcomes.

## Marijuana use and cognitive aptitude

The human brain is believed to be highly plastic and subject to considerable change from adolescence through young adulthood [16]. As such, the effects of substance use may be most significant during this period of the life course and it may affect the human brain in a variety of ways. A key focal area of research on marijuana and the developing adolescent brain has been on mental health. Studies have indicated that marijuana use may be associated with changes in the brain that lead to increased risk of mental health impairment. For example, Patton and colleagues [17] identified that young people who use marijuana may experience elevated symptoms of depression and anxiety. Marijuana use has also been associated with more significant psychiatric disorders, such as schizophrenia [18] and the onset of psychosis [19]. Although it has not been fully disentangled as to whether this is a causal relationship or if marijuana use occurs as a result of emergent mental health issues, the scholarly literature has hypothesized that this is one mechanism by which marijuana use affects the rapidly developing human brain.

Another concern regarding the use of marijuana and the human brain, particularly among young people, is the possibility that it affects cognitive processes and cognitive abilities in the long-term. Impairment of cognition while under the influence has been well established in the literature [7, 20–21]. At the same time, debates exist regarding the scope and duration of the effects of cannabis use on cognitive aptitude. The literature identifies that there are acute effects on cognition during intoxication, which can linger for days afterward [22]. However, these effects may not be durable over extended periods of time, particularly after the individual has been abstinent from use. Some research has indicated that cognitive deficits in memory function improve within one week of abstinence [23]. A recent meta-analysis about the effects of cannabis use on adolescents and young adults identified two major issues across the research literature: 1) many studies have found statistically significant results, but the magnitude of the effects may not lead to clinically relevant deficits, and 2) these effects may not persist after cessation of use [24]. Research on twin pairs has also indicated that the association between impaired cognitive aptitude and cannabis use may be attributable to familial characteristics that underlie both [25]. Such findings have renewed inquiries about selection into differential patterns of cannabis use.

Although there has been considerable inquiry into the possible cognitive effects of marijuana use, the influence of cognitive aptitude on selection into longitudinal patterns of substance use is less well understood. Recent research has called into question long-standing assumptions about the assumed causal direction that marijuana use impairs cognitive aptitude in the long-term [e.g. 25–26]. In contrast, this line of research has considered that young people may select into marijuana use on the basis of cognitive aptitude, and that the relationship is not necessarily a linear one between cognitive aptitude and use, nor in the assumed direction. White and Beatty [27] found that IQ at 10 years old was positively associated with greater odds of reporting past year cannabis use and cocaine use at age 30 after accounting for socioeconomic disadvantage and psychological distress. Similarly, Williams and Hagger-Johnson [28] indicated that children characterized as having high (vs. low) academic ability reported increased risk of occasional and persistent alcohol and cannabis use. Other work has identified

that academic ability in childhood, particularly math ability, is associated with greater odds of experiencing opportunities to try marijuana [29]. In this regard, the literature has begun to identify that cognitive abilities are associated with later patterns of substance use in adolescence and young adulthood.

Cognitive aptitude may not only play a role in immediate patterns of use and overall prevalence of whether or not individuals use, but may also shape trajectories of use over time as young people age into adulthood. Patterns of substance use over time can be highly heterogeneous and with differing implications for life course outcomes [12, 15]. As such, it remains important to account for varied patterns of use when considering whether cognitive aptitude may be associated with substance use as young people age from adolescence through young adulthood.

## Cognitive aptitude, sociality, and the role of peers in substance use

A number of explanations have been proffered for why cognitive aptitude may shape patterns of substance use over time. Cognitive aptitude may directly relate to health literacy and the ability to utilize information to achieve health and well-being [30]. Within the context of substance use, higher cognitive abilities among young people may shape health literacy and the utilization of information about the risks associated with substance use to produce reduced odds of use or less harmful patterns of use [27]. In this manner, we might expect that stronger cognitive aptitude would lead to lower substance use and fewer problems. Additionally, although shaped by socioeconomic background and parental cultural capital [31], cognitive aptitude is strongly associated with educational attainment [32]. Education provides a host of flexible resources to individuals, above and beyond individual cognitive aptitude, that shapes a range of health outcomes [33]. In this manner, educational trajectories and pathways established early on via cognitive abilities may situate young people for more positive health behaviors as they age into adulthood, including with respect to the use of substances such as marijuana.

Alternatively, the literature suggests that young people with higher cognitive abilities may be more likely to use substances, such as cannabis, because of a greater tendency towards novelty seeking and a desire to experiment [28]. In such instances, the drug experience may fulfill certain intellectual curiosities about experiences. Such curiosities may be greater in young people with higher cognitive abilities. On the whole then, the literature currently offers several psychological processes related to cognitive aptitude that may either enable or inhibit the use of cannabis among young people.

While the role of cognition as a selection mechanism has been considered, it has primarily been theorized as a psychological mechanism related to decision making [e.g. 34]. We contend, however, that cognitive factors may also play a role in how young people select into marijuana use through *social* processes, such as homophily, educational tracking, precocity, and strain. While patterns of selection driven by cognition may directly shape marijuana use, they also may be related to how peer groups are formed, which in turn shapes selection processes of substance use. Put simply, cognition may shape the peer networks in which young people find themselves [35]. These processes of social sorting related to cognition during early periods of adolescence may in turn lead to young people differentially nested within peer groups with varying levels of deviant activities such as smoking cigarettes or marijuana use. Such varying exposures to peers and deviant activities may in turn have an impact on trajectories of marijuana use from adolescence through young adulthood by establishing early patterns of behavior, social norms favorable to use, and exposure to opportunities to use.

Social sorting on the basis of cognition may occur due to general social network principles of homophily [36]. Children and adolescents develop and maintain friendships over time on the basis of shared attributes, aspirations, and activities, including substance use [37]. Yet, at its roots, homophily may partly emerge from shared cognitive similarity, with youth gravitating towards others who exhibit comparable interactional styles, interests, and goals that are dependent upon cognitive development. Patterns of cognitive development as youth age through the life course may create scenarios where young people cluster together within wider networks because they developed on similar cognitive trajectories and as a result generated interactional styles and interests on similar paths. These processes may lead to either reinforcement of psychological mechanisms through the group process–e.g. strengthening tendencies towards novelty seeking and experimentation within a high ability peer group–or create distinct social mechanisms by which cognitive aptitude is associated with substance use.

The educational tracking system may further structure social networks of young people according to cognitive aptitude because of how individuals are grouped together in classes according to educational achievement, particularly as young people reach grades 7 through 12. Educational tracking may increase the likelihood of forming friends of similar cognitive aptitude through mechanisms such as propinquity, similarity, and status [38–40]. As young people are grouped within many classes according to abilities, their peer networks may shift to reflect their time spent with peers whose abilities more similarly mirror their own. This in turn may shape how they become differentially exposed to opportunities to use substances as well as the potential for experiencing differing social norms. In this manner, the educational tracking system also may contribute to the clustering of peers with varying levels of deviance [41] and in turn shape norms, activities, and opportunities related to substance use.

The relationship of cognitive aptitude to precocious transitions may result in differential uptake of substance use among youth. Precocious transitions for young people involve taking on behaviors, including substance use, related to roles and responsibilities without sufficient preparation [42–43]. Early pubertal development has been associated with uptake of drinking and smoking among adolescents [44]. Adolescents with higher cognitive abilities may be more likely to strive for adult roles and responsibilities while still in adolescence, and may be more likely to take up substance use as a result. Additionally, adolescents with precocious transitions stemming from cognitive aptitude may be more likely to develop relationships with older peers, and in turn be socialized into substance use by the more advanced peers.

Cognitive aptitude may also shape patterns of friendship formation driven by socially produced strains as described by Agnew's [45] General Strain Theory. First, individuals who are either high or low in cognitive aptitude may be more susceptible to bullying within wider youth networks and school settings, introducing the strain of negative stimuli articulated by Agnew. These youth may then cluster together on the basis of shared experiences of isolation and disapproval from other classmates. For example, prior research has identified that youth cluster together on the basis of bullying tendencies [46]. Youth are more likely to increase bullying behavior over time when peers they admire engage in bullying, net of generalized popularity [47], which may amplify the network clustering. Second, with respect to lower cognitive aptitude, the process may emerge due to the domain of failure to achieve positively valued goals as a source of strain. The strains of weaker scholastic performance produced within family and school contexts may lead to peer deviant selection or rejection by stably performing peers. Substance use among adolescents may also be a means of coping with strain generated on the basis of cognitive abilities, both on the individual level as well as within peer networks of youth who share that source of strain.

In sum, there are a number of reasons to expect that cognitive aptitude may shape peer networks and in turn divergent peer group structures may lead to differential patterns of

substance use among young people. We may therefore question: if there are selection effects into marijuana use, are they robust to controlling for peer group behaviors?

## Current study

Given the role of cognition in shaping patterns of substance use identified in the literature, we first aim to examine the relationship of cognitive aptitude in early adolescence to patterns of marijuana use over the course of the transition from adolescence to adulthood within a national sample of youth. Importantly, we examine heterogeneous trajectories of marijuana use that move beyond merely the presence or absence of use to account for varying frequency over time. In this manner, we extend the discussion of the relationship of cognition to marijuana use beyond lifetime use to account for the complexity of patterns of consumption over time. The literature reviewed above provides evidence for the possibility of two competing hypotheses.

As noted above, the literature indicates that higher cognitive abilities may inhibit substance use due to greater health literacy and ability to recognize risks associated with substance use [30]. As such, we may posit:

$H_{1A}$: *Lower cognitive aptitude will lead to heavier trajectories of marijuana use from adolescence through young adulthood.*

Additionally, the literature has indicated that higher cognitive aptitude may increase propensity to use marijuana due to heightened curiosity and desire for sensation seeking among high aptitude young people [28]. As such, we may also conjecture:

$H_{1B}$: *Higher cognitive aptitude will lead to heavier trajectories of marijuana use from adolescence through young adulthood.*

Moving beyond the individual, we also described several social mechanisms by which cognitive aptitude may shape trajectories of marijuana use. Given that cognition may organize how peer groups are formed and maintained, the possibility exists that peer group deviance might explain what appears to be selection based on cognition. Much like the main effect hypotheses, the literature provides for the possibility of two competing hypotheses on how cognition relates to peers and marijuana use trajectories. These relationships are reinforced by general principles of homophily as well as the educational tracking system in terms of how young people select into friendships.

Extending from theories of general strain, lower cognitive aptitude may lead to selection into deviant friendship networks that facilitate substance use behaviors. As such, we may posit:

$H_{2A}$: *Lower cognitive aptitude will lead to peer group formation that facilitates heavier trajectories of marijuana use from adolescence through young adulthood.*

Alternatively, aligning with theories on precocious adolescent behavior, higher cognitive aptitude may lead to precocious transitions that place young people within peer groups that facilitate substance use. In this manner, we may consider that:

$H_{2B}$: *Higher cognitive aptitude will lead to peer group formation that facilitates heavier trajectories of marijuana use from adolescence through young adulthood.*

The current study moves the literature forward regarding the relationship of early adolescent cognition to marijuana use over time, as well as the role of peers in this association, in the following way. First, we describe heterogeneous trajectories of marijuana use from ages 16 to 26 in a nationally representative sample. Then, we identify whether young people select into certain trajectories of marijuana use based on cognitive aptitude measured during early

adolescence. Finally, we assess whether these effects (if any) are robust to the inclusion of measures of peer group deviance. Overall, we are able to leverage a national sample of young people making the transition to adulthood to examine whether cognition matters for trajectories of marijuana use and whether peers play a role in this process.

## Methods

### Data

The data come from the National Longitudinal Survey of Youth 1997 (NLSY97). The NLSY97 is a large, nationally representative sample (*N* = 8,984) designed to track the transition of youth into adulthood. Adolescents ages 12 to 16 were randomly sampled in 1997 and surveyed annually through 2011. Within our models, we utilize adolescents who were ages 12 to 15 at baseline so that we may establish temporal direction prior to the trajectories of marijuana use, which are modeled from age 16 to 26 (see below). The retention rate was about 80 percent by 2011. In a longitudinal dataset, researchers must choose between year and age as the time metric based upon theoretical considerations [48]. Given that age is central to substance use among adolescents and young adults, we chose age as our metric for the latent trajectory analysis.

**Marijuana use.** Our main variable of interest is marijuana use, assessed in each survey. Overall, marijuana use among the NLSY97 cohort coheres with other national surveys. We are interested in the frequency of marijuana use over time, as opposed to simply any use. Therefore, we utilize the question: "On how many days have you used marijuana in the last 30 days?" Using the categories 0 days, 1 to 2 days, 3–9 days, 10–24 days, and 25–30 days, we previously identified heterogeneous pathways of marijuana use from age 16 to 26 [15; see Fig 1 of the latent trajectory models). In our consideration of marijuana use, we began the age-based trajectories at age 16, the youngest age at which all cohorts contribute data; marijuana prior to age 16 is very low–the first peak in the hazard for initiation occurs at age 16 [49]. These extend through age 26, the oldest age at which all cohorts have consecutive annual data. This latent trajectory analysis identified five trajectories over time shown in Fig 1: *Abstainers*, whose most typical response was to not have used marijuana in the prior 30 days over time; *Dabblers*, who had a high probability of low use during the prior 30 days; *Consistent users* regularly used marijuana at mixed levels throughout the period of observation. *Early heavy quitters* reported heavy use during the late teenage years, but virtually no use from age 22 onward; *Persistent heavy users* reported steady heavy use across the period of observation. The model resulting in the identification of these trajectories was selected on the basis of appropriate fit indices for latent trajectory analysis, such as Bayesian Information Criterion.

**Cognition.** The Armed Services Vocational Aptitude Battery (ASVAB) Math-Verbal percentile is an age-adjusted score that measures cognitive aptitude in domains such as mathematical knowledge, arithmetic reasoning, word knowledge, and paragraph comprehension. The measure was designed for use with high school students, and has verified content validity and construct validity [50]. It measures cognitive aptitude in a manner that is predictive of performance on occupational tasks. It is strongly correlated with other measures of cognitive aptitude such as the SAT, ACT, and numerous other standard intelligence tests [51]. The ASVAB is a commonly used predictor in a wide range of studies [e.g.52–54]. For a more meaningful interpretation of differences between individuals on cognition, we divided the variable by 10, permitting us to interpret the measure as a decile increase rather than the small magnitude associated with an increase of a percentile.

**Peer influence.** To assess the role of peers in these processes, we included two measures. The first is the proportion of peers at baseline who use illegal drugs, a measure that includes

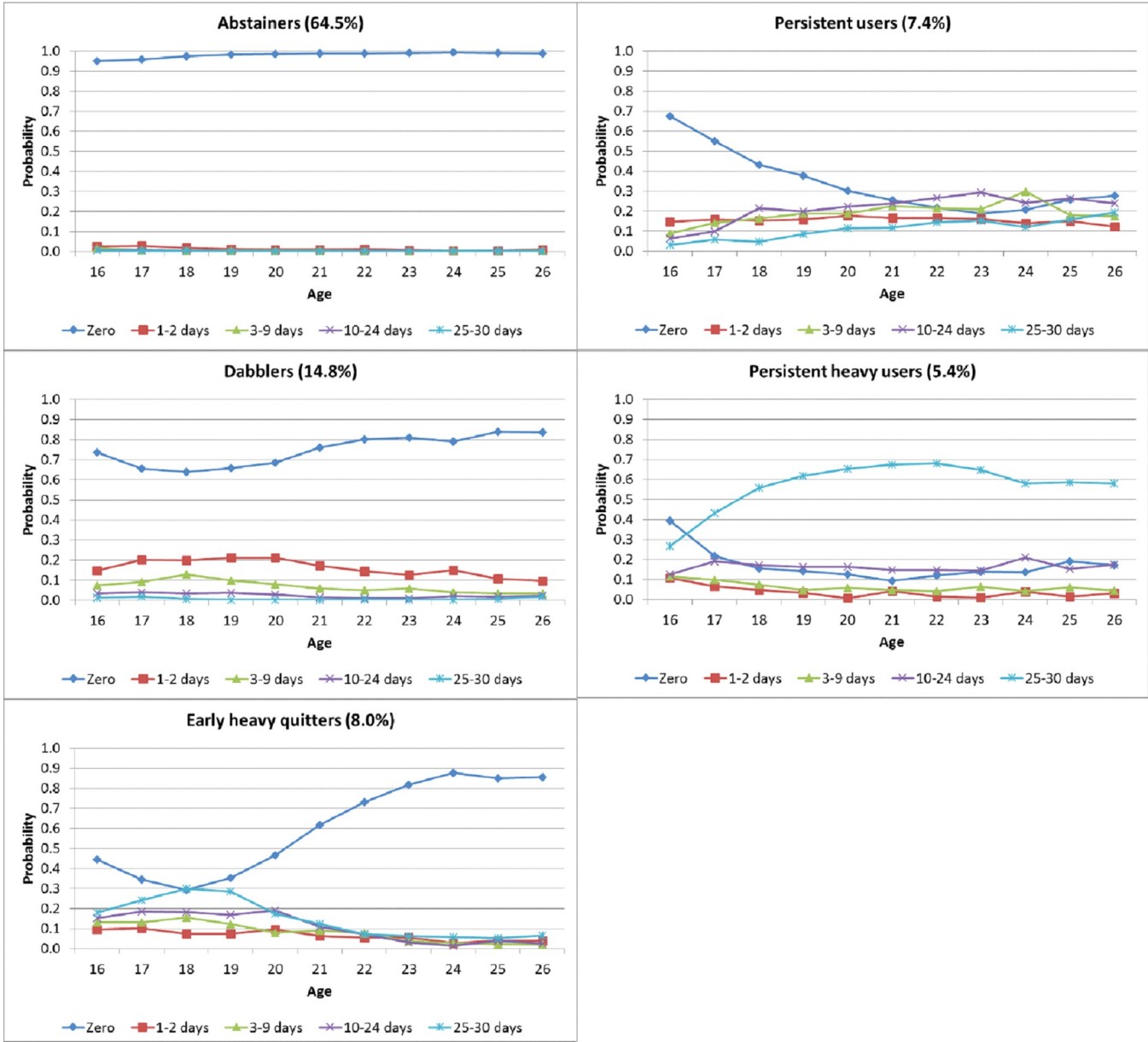

*Source*: Kelly and Vuolo (2018).

*Note*: Percentages in figures correspond to the representation from the latent trajectory model of the whole NLSY97 sample, including 16 year olds. For the percentages for the subsample of 12-15 year olds used in this manuscript, see Table 1.

**Fig 1. Latent trajectory analyses of marijuana use over time.** *Source*: Kelly and Vuolo (2018).

marijuana. Also, given that the prevalence of illegal drug use (even including marijuana) is low among youth 12–15 years of age, in a robustness check we included the proportion of peers at baseline who smoke cigarettes. This permits us to assess the model with a substance that has typically been used much earlier in adolescence, a reason for which cigarettes have been

characterized as a gateway drug. Both measures were five category ordinal measures ranging from "almost none–less than 10%" to "almost all-more than 90%."

**Control variables.** Demographics assessed include gender, race/ethnicity, and employment at age 16 (the start of the trajectory), as well as a control for age cohort. We include covariates for parents' highest education, which is a proxy for socioeconomic background, and parent's self-rated health. Finally, given the strong linkages between mental health and substance use, we control for depression using the five-item short version of the Mental Health Inventory (MHI-5). We do not include educational performance because the ASVAB has been shown to be so highly correlated with school performance that their effects cannot be separated [55]. Table 1 shows descriptive statistics for all variables, including the percentages among the marijuana trajectories.

## Analysis

We used logistic regression analysis to assess the role of cognition on any marijuana use by age 26. Subsequently, we utilized multinomial logistic regression to analyze the effects of cognitive aptitude on trajectories of marijuana use.

**Table 1. Demographic characteristics of NLSY subjects under age 16.**

|  | Mean (SD) / % |
|---|---|
| Cognitive Aptitude | 45.46 (29.15) |
| *Marijuana Trajectories* |  |
| Abstainers | 66.40% |
| Dabblers | 13.25% |
| Consistent Users | 7.78% |
| Persistent Heavy Users | 5.50% |
| Early Heavy Quitters | 7.07% |
| Depression | 4.65 (2.55) |
| Age at baseline | 13.38 (1.07) |
| Female | 48.61% |
| *Parental education* |  |
| Less than High School Degree | 15.84% |
| High School Degree | 32.47% |
| Some College | 25.53% |
| 4 years of College or more | 26.16% |
| *Race/Ethnicity* |  |
| White | 50.08% |
| Black or African American | 26.23% |
| American Indian, Eskimo, Aleut | 0.76% |
| Asian or Pacific Islander | 1.72% |
| Other | 1.38% |
| Hispanic or Latino | 19.83% |
| *Parental Health* |  |
| Good-Excellent | 76.02% |
| Fair-Poor | 12.99% |
| No parent info | 11.00% |
| *Employment Status* |  |
| Not working | 65.36% |
| Part-time | 30.96% |
| Full-time | 3.69% |

## Results

We first provide a multivariable model that assesses the relationship of cognitive aptitude to lifetime use of marijuana by early adulthood in order to emphasize the importance of accounting for heterogeneous patterns of use over time, as described in later models. The analyses presented in Table 2 demonstrate the lack of association when examining the role of early adolescent cognition on lifetime marijuana use. As indicated in the multivariable logistic model, there is no relationship between cognitive aptitude and the odds of any lifetime marijuana use by age 26. Thus, cognitive aptitude does not account for merely the presence or absence of marijuana use by early adulthood. However, as the analyses in Table 3 show, accounting for heterogeneous trajectories of marijuana use from adolescent through young adulthood yields some interesting results.

Within Table 3, we present three multinomial models that allow us to examine the hypotheses outlined earlier. We examine the relative risk of entry into each trajectory compared to the baseline of the abstainer trajectory. Model 1 provides the bivariate relationship between cognition and the trajectory, Model 2 introduces all of the covariates identified above, and Model 3 introduces the peer deviance measure to assess whether the effect changes with the introduction of peer substance use. Although we only present the variables of interest within Table 3, a full table of the models with the covariates is available in S1 Table. In the bivariate models, we see that, relative to the non-using trajectory, one decile higher in cognitive aptitude in early

**Table 2.** Logistic regression of any use of marijuana by age 26 (n = 4,231).

| | Adjusted Odds Ratio (95% CI) |
|---|---|
| Cognition | 1.012 (0.986–1.039) |
| Depression | 1.139*** (1.109–1.170) |
| Age at baseline (Ref: 12) | |
| 13 | 0.881 (0.743–1.046) |
| 14 | 0.915 (0.770–1.087) |
| 15 | 0.952 (0.788–1.151) |
| Female | 0.668*** (0.587–0.761) |
| Parental education (Ref: <HS degree) | |
| High School Degree | 1.021 (0.826–1.262) |
| Some College | 1.422 (1.133–1.784) |
| 4 years of College or more | 1.228 (0.966–1.560) |
| Race/Ethnicity (Ref: White) | |
| Black or African American | 0.715*** (0.606–0.843) |
| American Indian, Eskimo, Aleut | 0.770 (0.389–1.523) |
| Asian or Pacific Islander | 0.559* (0.340–0.919) |
| Other | 1.443 (0.818–2.547) |
| Hispanic or Latino | 0.928 (0.768–1.122) |
| Parental Health (Ref: Excellent-Good) | |
| Fair-Poor | 1.164 (0.954–1.421) |
| No parent info | 1.324* (1.029–1.705) |
| Employment Status (Ref: Unemployed) | |
| Part-time | 1.029 (0.898–1.179) |
| Full-time | 1.793*** (1.225–2.625) |

*$p < .05$; **$p < .01$

***$p < .001$

**Table 3. Multinomial logistic regression of cognitive aptitude and heterogeneous trajectories of marijuana use from age 16 through age 26 (n = 3,903).**

| | Dabblers | | |
| --- | --- | --- | --- |
| | Model 1 | Model 2 | Model 3 |
| | Relative Risk (95% CI) | Relative Risk (95% CI) | Relative Risk (95% CI) |
| Cognition | 1.083*** (1.049–1.118) | 1.048* (1.007–1.090) | 1.054** (1.013–1.097) |
| Peers Drug Use | | | |
| About 25% | | | 1.410** (1.098–1.810) |
| About half (50%) | | | 1.934*** (1.457–2.569) |
| About 75% | | | 1.918*** (1.347–2.732) |
| Almost all (more than 90%) | | | 1.626* (1.048–2.524) |
| | Consistent Users | | |
| | Model 1 | Model 2 | Model 3 |
| | Relative Risk (95% CI) | Relative Risk (95% CI) | Relative Risk (95% CI) |
| Cognition | 1.123*** (1.075–1.173) | 1.126*** (1.066–1.189) | 1.134*** (1.074–1.199) |
| Peers Drug Use | | | |
| About 25% | | | 1.806** (1.290–2.528) |
| About half (50%) | | | 1.938** (1.279–2.937) |
| About 75% | | | 2.537*** (1.572–4.096) |
| Almost all (more than 90%) | | | 1.928* (1.016–3.660) |
| | Persistent Heavy Users | | |
| | Model 1 | Model 2 | Model 3 |
| | Relative Risk (95% CI) | Relative Risk (95% CI) | Relative Risk (95% CI) |
| Cognition | 1.047 (0.999–1.096) | 1.030 (0.972–1.091) | 1.052 (0.992–1.115) |
| Peers Drug Use | | | |
| About 25% | | | 1.756** (1.169–2.636) |
| About half (50%) | | | 3.471*** (2.273–5.300) |
| About 75% | | | 5.460*** (3.430–8.691) |
| Almost all (more than 90%) | | | 4.274*** (2.403–7.605) |
| | Early Heavy Quitters | | |
| | Model 1 | Model 2 | Model 3 |
| | Relative Risk (95% CI) | Relative Risk (95% CI) | Relative Risk (95% CI) |
| Cognition | 0.963 (0.924–1.003) | 0.917** (0.871–0.965) | 0.937* (0.889–0.987) |
| Peers Drug Use | | | |
| About 25% | | | 1.454* (1.026–2.061) |
| About half (50%) | | | 2.130*** (1.445–3.139) |
| About 75% | | | 3.752*** (2.472–5.696) |
| Almost all (more than 90%) | | | 4.704*** (3.014–7.344 |

*p < .05

**p < .01

***p < .001

Note: Reference outcome category is Abstainers. Model 1 = bivariate model. Model 2 introduces all covariates. Model 3 introduces peer drug use. See Supporting Information (S1 Table) for results with all covariates.

adolescence is associated with 8.3% greater relative risk of the dabbler trajectory (RR = 1.083; p < .001) and 12.3% higher relative risk of the consistent user trajectory (RR = 1.123; p < .001). No significant relationship with the persistent heavy user trajectory or early heavy quitter trajectory was found. With the introduction of covariates into Model 2, we see similar results. For each decile increase in cognition and compared to abstainers, the likelihood of

**Table 4. Cognitive aptitude and heterogeneous trajectories of marijuana use from age 16 through age 26, using peer smoking (n = 3,948).**

| | Dabblers | Consistent Users | Persistent Heavy Users | Early Heavy Quitters |
|---|---|---|---|---|
| | Relative Risk (95% CI) | Relative Risk (95% CI) | Relative Risk (95% CI) | Relative Risk (95% CI) |
| Cognition | 1.050* (1.009–1.092) | 1.140*** (1.079–1.205) | 1.053 (0.993–1.117) | 0.931** (0.884–0.980) |
| Peer Smoking | | | | |
| About 25% | 1.089 (0.838–1.414) | 1.758** (1.230–2.512) | 1.629* (1.078–2.462) | 1.543* (1.079–2.207) |
| About half (50%) | 1.386 (1.049–1.830) | 2.458*** (1.674–3.608) | 2.339*** (1.517–3.607) | 2.147*** (1.481–3.112) |
| About 75% | 1.472* (1.081–2.004) | 2.245*** (1.434–3.516) | 2.875*** (1.805–4.579) | 2.570*** (1.729–3.821) |
| Almost all (more than 90%) | 1.084 (0.694–1.695) | 1.939* (1.029–3.655) | 3.447*** (1.939–6.130) | 2.186** (1.295–3.689) |

*p < .05

**p < .01

***p < .001

Note: Model inclusive of all covariates. Reference outcome is Abstainers.

being in the dabbler trajectory increases by 4.8 percent (p < .05) and of being in the consistent user trajectory by 12.6 percent (p < .001), and now a 8.3 percent lower relative risk for the early heavy quitter trajectory (p < .01). To further demonstrate the impact of cognition, we consider the distinction for a difference of 5 deciles. For example, we can consider youth across the middle two quartiles at the 25[th] percentile versus those at the 75[th] percentile. For such a distinction between these young people, the relative risk of entry into the dabblers group is 26.5% greater and entry into the consistent users group is 81.3% greater for the youth scoring at the 75[th] percentile than youth scoring at the 25[th] percentile. Thus, there are notable distinctions for a gap of this size, even among youth scoring in the middle ranges of such cognitive aptitude tests.

With the introduction of the peer illegal drug use variable in model three, the results are virtually unchanged. There is little influence on the relationship between cognitive aptitude and relative risk of marijuana trajectories, suggesting that peer substance use patterns are not mediating this relationship. As a robustness check, comparable models were run using a measure of peer group cigarette smoking rather than illegal drug use. Again, we find substantively similar findings and the effects are virtually unchanged. Although not presented here, these analyses are available in Table 4.

Table 5 provides the results of alternative base comparisons made in the course of the multinomial analyses. Two major comparisons beyond those made to the abstainer trajectory stands out. First, in comparison to the consistent user trajectory, the results displayed in Table 5 indicate that each decile increase in cognitive aptitude in early adolescence lowers the relative risk of placement in three of the other trajectories, including 11.9% lower risk of being abstainers (RR = 0.881; p < .001), 7.1% lower risk of dabblers (RR = 0.929; p < .05), and 17.4% lower risk for early heavy quitters (RR = 0.826; p < .001). The exception is persistent heavy users relative to consistent users, which closely approaches statistical significance (RR = 0.927; p = .052). Second, individuals with higher cognitive aptitude in early adolescence have a higher relative risk of placement in any of the other trajectories besides the early heavy quitters, including abstainers (RR = 1.068; p < .05), dabblers (RR = 1.126; p < .001), consistent users (RR = 1.211; p < .001), and persistent heavy users (RR = 1.123; p < .01). The relationship for all of the above results remained statistically significant in the models that included peer group smoking (see Table 6). Overall, for higher cognitive aptitude youth, the relative risk of becoming a consistent user was greater than for any other possibility, while the relative risk of becoming an early heavy quitter was lower than any other possibility.

**Table 5. Key comparisons for other base groups (n = 3,903).**

| | Base as Consistent Users | Base as Early Heavy Quitters |
|---|---|---|
| | Relative Risk (95% CI) | Relative Risk (95% CI) |
| Abstainers | 0.881*** (0.834–0.931) | 1.068* (1.014–1.125) |
| Dabblers | 0.929* (0.872–0.990) | 1.126*** (1.059–1.196) |
| Consistent Users | Ref. | 1.211*** (1.128–1.301) |
| Early Heavy Quitters | 0.826*** (0.769–0.887) | Ref. |
| Persistent Heavy Users | 0.927 (0.859–1.001) | 1.123** (1.043–1.209) |

*p < .05
**p < .01
***p < .001
Note: Full model with peer drug use and all covariates. Model without peer drug use produced comparable results.

## Discussion

Historically, the relationship between marijuana use and cognition has been studied from the perspective of assessing how marijuana use affects cognition. Given the potential of cognition to shape patterns of substance use identified more recently in the literature, we set out to examine the relationship of cognitive aptitude in early adolescence to trajectories of marijuana use over the course of the transition from adolescence to adulthood. Importantly, we moved beyond simple dichotomous assessments of lifetime marijuana use to examine heterogeneous trajectories of marijuana use over time. Overall, the results indicate that while cognition appears to have no association with any lifetime marijuana use by age 26, a relationship emerges when considering the varying trajectories of marijuana consumption over this important transitional period of the life course.

We presented two sets of hypotheses regarding the relationship of marijuana use based upon the extant literature. Our first hypothesis specified that lower cognitive aptitude will lead to heavier trajectories of marijuana use from adolescence through young adulthood. The existing literature also suggested an alternative hypothesis that higher cognitive aptitude will lead to heavier trajectories of marijuana use from adolescence through young adulthood. The results of our analyses suggest a more complicated set of findings that point towards the nuances of extended periods of marijuana use. Instead, we have identified that this relationship is complex and nonlinear.

**Table 6. Key comparisons for other base groups, using peer smoking (n = 3,948).**

| | Base as Consistent Users | Base as Early Heavy Quitters |
|---|---|---|
| | Relative Risk (95% CI) | Relative Risk (95% CI) |
| Abstainers | 0.877*** (0.830–0.926) | 1.075** (1.020–1.132) |
| Dabblers | 0.921** (0.864–0.981) | 1.128*** (1.062–1.199) |
| Consistent Users | Ref. | 1.225*** (1.141–1.316) |
| Early Heavy Quitters | 0.816*** (0.760–0.876) | Ref. |
| Persistent Heavy Users | 0.924* (0.856–0.997) | 1.132** (1.052–1.219) |

*p < .05
**p < .01
***p < .001
Note: Full model with peer smoking and all covariates. Model without peer smoking produced comparable results.

Complicating our stated hypotheses, rather than simply cognition being associated with heavier or lighter patterns of use, we find that higher cognitive aptitude adolescents have greater relative risk of falling into trajectories of becoming regular yet measured marijuana users as they transition from middle adolescence into young adulthood. In this regard, they seem to become more likely to use marijuana over time but also exhibit a degree of control that prevents them from progressing to heavy extended patterns of use. In this manner, aspects of sensation seeking tied to cognitive aptitude may facilitate entry into use [28], and yet countervailing cognitive factors related to health literacy [30] and flexible resources related to education [33] may also prevent patterns of use from escalating to the most severe forms of cannabis consumption. As such, while higher cognitive aptitude youth may engage in regular marijuana use, they may also minimize the probability of experiencing the most serious consequences of marijuana use as they age from adolescence into adulthood [12–15].

In addition to considering the relationship of cognition to patterns of marijuana use, we were also interested in the role that peer group deviance may play in mediating the relationship between cognitive aptitude and trajectories of marijuana use. We considered several conceptual frameworks–homophily, educational tracking, precocity, and strain–with implications for the possibility that cognition may organize how peer groups are formed and maintained, which in turn shapes marijuana use trajectories into young adulthood. Given the lack of change to the magnitude of the coefficients or their levels of significance, the results suggest that peer substance use does not mediate the relationship between cognition and trajectories of marijuana use. In other words, it appears unlikely that cognition is shaping patterns of substance use because it leads youth to select into peer networks that facilitate substance use. Yet peers clearly matter as well, such that cognition and peers *independently* contribute to trajectories of marijuana use in the early life course.

The identification of factors that shape patterns of substance use are important with respect to the development of prevention, education, and outreach efforts among young people. Regarding the implications of our findings for these efforts, our results indicate that youth who exhibit higher cognitive aptitude may benefit from additional attention through prevention and health intervention efforts. Yet, we also note that since they seem to avoid the heaviest trajectories of marijuana use already, they may be well equipped with respect to health literacy in ways that benefit them with respect to information that may help them avoid the most significant problems associated with extended heavy marijuana use. This is an area of research that requires deeper study. Yet, evidence-based approaches to preventing harms from substance use may be especially well suited for young people with high cognitive aptitude.

## Limitations

While these results provide important additional information on the role of cognitive aptitude in patterns of marijuana use over time, we must consider some limitations. First, the design does not permit assertions of causality. While we included numerous important covariates, causality cannot be fully inferred from this observational data using the models employed. Additionally, although we assess past month marijuana use in the trajectories, patterns of use could differ during other months; yet, we do not anticipate that such patterns would cause respondents to resemble those of a different trajectory. Still, there is use not captured in the trajectories, particularly for abstainers. Within the NLSY97, 58.8 percent of the respondents had ever tried marijuana. In fact, some of those in the abstainers trajectory have tried marijuana at some point although their use does not register in the data, indicating that they may have experimented merely once or twice and did not develop regular patterns of use. Yet, it makes sense that the youth who tried marijuana only once or twice in their lives would be

classified alongside those who never used rather than with those who use marijuana a few occasions per year over time; their lives and social activities are likely to be more similar to "true abstainers" than those who continue to use marijuana over time, even at lower levels [see 15 for further information]. Furthermore, although a widely used and comprehensive measure of cognitive aptitude, we recognize that the ASVAB is not the only means by which cognitive aptitude may be assessed. Alternative assessments should be pursued in future research.

In addition, we acknowledge that although peer substance use has widely been documented as shaping individual substance use, peer group deviance may not be the only mechanism by which peers may shape processes of selection into marijuana use. Other aspects of peer relationships should be investigated in future research, including issues of social support, popularity, and subcultural involvement. Furthermore, we recognize that peer groups change over time and this likely shapes (and is shaped by) these trajectories, but the NLSY does not contain annual measures of peer group deviance. Future work on the relationship between cognition, peers, and substance use would benefit from analytic opportunities offered by repeated measures of all three, such as through cross-lagged panel models. Similarly, depression is not measured annually across time.

Finally, we recognize that NLSY97 respondents came of age prior to considerable legal change surrounding marijuana. Not only has medical and recreational marijuana been increasingly adopted by states, but public opinion on legalization and acceptability has undergone large shifts as well [56]. In particular, daily marijuana use for young adults has steadily increased recently, particularly among the most regular users. [57]. Thus, among those in the persistent heavy user and early heavy quitter trajectories, there could be even heavier use than that among the NSLY97 respondents. Yet, there are also similarities between the two cohorts. Despite legalization, perceptions of marijuana as easy to acquire have remained similar among high school seniors (and in fact, is slightly *lower* among recent cohorts); that is, high school seniors have always perceived marijuana as easy to access [58]. Although daily rates of young adult use have increased, rates corresponding to the NLSY97 cohort of any and daily use of marijuana were quite similar to current cohorts, and in fact were higher than many cohorts in-between [58]. Simultaneously, the number of new teenage initiates has significantly decreased [57]. Thus, the existing evidence indicates that increase in frequency of use may occur primarily among those inclined to be among the heavier users, and such within-trajectory change would not appreciably change the findings about the trajectories. Whether this would alter the percentages in the marijuana trajectory groups is unknown without the availability of data from more recent cohorts. Taking these considerations together, there is little reason to believe that cohort differences would alter the underlying relationship between cognition prior to marijuana use and trajectories in young adulthood, but this should be confirmed empirically. There are unfortunately few national studies with panel designs and annually repeated measures of marijuana use that track transitions from adolescence to young adulthood, which requires an extended period of observation. The NLSY97 remains one of the best nationally representative longitudinal with which to examine this research question.

## Conclusions

The results of our study indicate that adolescents in a national sample who rate higher in cognitive aptitude during early adolescence may be more likely to enter into consistent but not extreme trajectories of marijuana use as they age into young adulthood. At the same time, they are less likely to enter into patterns of heavy use during adolescence and desist by early adulthood. The role of cognition as a selection mechanism into marijuana use has taken on heightened importance. Yet, cognition may not influence patterns of marijuana use over time by

organizing peer groups. As policies regarding marijuana use are in a period of flux within the United States and other nations, the identification of how people enter into various patterns of marijuana use has increased in significance, and these processes merit continued monitoring in the changing policy environment.

## Supporting information

**S1 Table. Cognitive aptitude and heterogeneous trajectories of marijuana use with inclusion of peer drug use: Full models (n = 3,903) .**
(DOCX)

## Author Contributions

**Conceptualization:** Brian C. Kelly, Mike Vuolo.

**Formal analysis:** Brian C. Kelly, Mike Vuolo.

**Investigation:** Brian C. Kelly, Mike Vuolo.

**Methodology:** Brian C. Kelly, Mike Vuolo.

**Supervision:** Brian C. Kelly, Mike Vuolo.

**Writing – original draft:** Brian C. Kelly, Mike Vuolo.

**Writing – review & editing:** Brian C. Kelly, Mike Vuolo.

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
