## [Decision Letter · Decision Letter 0]

4 Sep 2019

[EXSCINDED]

PONE-D-19-16918

Cognitive Aptitude, Peers, and Trajectories of Marijuana Use from Adolescence through Young Adulthood

PLOS ONE

Dear Dr Kelly,

Thank you for submitting your manuscript to PLOS ONE. After careful consideration, we feel that it has merit but does not fully meet PLOS ONE’s publication criteria as it currently stands. Therefore, we invite you to submit a revised version of the manuscript that addresses the points raised during the review process.

Please complete all changes suggested by the reviewer below. We believe your article has merit and warrants publication in PLOS ONE if the revisions are made.

We would appreciate receiving your revised manuscript by Oct 19 2019 11:59PM. To enhance the reproducibility of your results, we recommend that if applicable you deposit your laboratory protocols in protocols.io, where a protocol can be assigned its own identifier (DOI) such that it can be cited independently in the future. For instructions see: http://journals.plos.org/plosone/s/submission-guidelines#loc-laboratory-protocols

We look forward to receiving your revised manuscript.

Kind regards,

Samantha S. Goldfarb, DrPH

Academic Editor

PLOS ONE

Journal Requirements:

1. Please ensure that you refer to Figures Appendix A in your text as, if accepted, production will need this reference to link the reader to the figure.

2. Please upload a copy of Figures 1 & 2, to which you refer in your text on page xx. If the figure is no longer to be included as part of the submission please remove all reference to it within the text.

Additional Editor Comments (if provided):

Thank you for submitting your article to PLOS ONE. We ask that you address the reviewer's concerns below and resubmit for potential publication.

Reviewers' comments:

Reviewer's Responses to Questions

**Comments to the Author**

1. Is the manuscript technically sound, and do the data support the conclusions?

Reviewer #1: Yes

2. Has the statistical analysis been performed appropriately and rigorously? 

Reviewer #1: Yes

3. Have the authors made all data underlying the findings in their manuscript fully available?

Reviewer #1: Yes

4. Is the manuscript presented in an intelligible fashion and written in standard English?

Reviewer #1: Yes

5. Review Comments to the Author

Reviewer #1: The focus of this manuscript is the potential impact of cognitive aptitude during adolescence on trajectories of marijuana use through age 26. The manuscript addresses a timely health behavior, presents a clear argument, and is well-written. While there are many strengths, there are also some major and minor concerns.

Major

The dependent variable for the analyses presented in this manuscript are latent classes of marijuana use based on trajectories over a ten-year period. However, all independent variables are from the baseline data. While this is less of a concern for the more variables more likely to be fixed (e.g., basic demographics), it is reasonable to expect that peer groups from age 16 to 26 would be very dynamic. Was peer use assessed at the multiple data collection time points? Can it be examined as a time-varying predictor? Similar comment for depression.

The most recent year data was collected is 2011 – almost 9 years old. What evidence suggests that the findings from this cohort would generalize to the current generation of adolescents transitioning into adulthood?

Minor

Page 16, Line 365. The text states the “analyses below” in reference to “Table 2”. You may want to consider referencing “Table 3” specifically rather than “below” for clarity.

Page 21, Line 469. Given that higher cognitive aptitude was associated with consistent use, it would be better to use “heavier” rather than “long-term”.

Please include sample sizes and referent groups for all tables.

6. PLOS authors have the option to publish the peer review history of their article (what does this mean?). If published, this will include your full peer review and any attached files.

Reviewer #1: No

---

## [Author Response · Author response to Decision Letter 0]

6 Sep 2019

we have included a file with our reply to the comments

---

## [Editor Report · Decision Letter 1]

16 Sep 2019

Cognitive Aptitude, Peers, and Trajectories of Marijuana Use from Adolescence through Young Adulthood

PONE-D-19-16918R1

Dear Dr. Kelly,

We are pleased to inform you that your manuscript has been judged scientifically suitable for publication and will be formally accepted for publication once it complies with all outstanding technical requirements.

With kind regards,

Samantha S. Goldfarb, DrPH

Academic Editor

PLOS ONE
---

## [Editor Report · Acceptance letter]

16 Oct 2019

PONE-D-19-16918R1 

Cognitive Aptitude, Peers, and Trajectories of Marijuana Use from Adolescence through Young Adulthood 

Dear Dr. Kelly:

I am pleased to inform you that your manuscript has been deemed suitable for publication in PLOS ONE. Congratulations! Your manuscript is now with our production department. 

With kind regards,

on behalf of

Dr. Samantha S. Goldfarb 

Academic Editor

PLOS ONE